# Algorithms and Techniques for the Structural Health Monitoring of Bridges: Systematic Literature Review

**DOI:** 10.3390/s23094230

**Published:** 2023-04-24

**Authors:** Omar S. Sonbul, Muhammad Rashid

**Affiliations:** Computer Engineering Department, Umm Al Qura University, Makkah 21955, Saudi Arabia; mfelahi@uqu.edu.sa

**Keywords:** structural health monitoring, bridges, machine learning, pattern recognition, feature extraction, systematic literature review, ANN, CNN

## Abstract

Structural health monitoring (SHM) systems are used to analyze the health of infrastructures such as bridges, using data from various types of sensors. While SHM systems consist of various stages, feature extraction and pattern recognition steps are the most important. Consequently, signal processing techniques in the feature extraction stage and machine learning algorithms in the pattern recognition stage play an effective role in analyzing the health of bridges. In other words, there exists a plethora of signal processing techniques and machine learning algorithms, and the selection of the appropriate technique/algorithm is guided by the limitations of each technique/algorithm. The selection also depends on the requirements of SHM in terms of damage identification level and operating conditions. This has provided the motivation to conduct a Systematic literature review (SLR) of feature extraction techniques and pattern recognition algorithms for the structural health monitoring of bridges. The existing literature reviews describe the current trends in the field with different focus aspects. However, a systematic literature review that presents an in-depth comparative study of different applications of machine learning algorithms in the field of SHM of bridges does not exist. Furthermore, there is a lack of analytical studies that investigate the SHM systems in terms of several design considerations including feature extraction techniques, analytical approaches (classification/ regression), operational functionality levels (diagnosis/prognosis) and system implementation techniques (data-driven/model-based). Consequently, this paper identifies 45 recent research practices (during 2016–2023), pertaining to feature extraction techniques and pattern recognition algorithms in SHM for bridges through an SLR process. First, the identified research studies are classified into three different categories: supervised learning algorithms, neural networks and a combination of both. Subsequently, an in-depth analysis of various machine learning algorithms is performed in each category. Moreover, the analysis of selected research studies (total = 45) in terms of feature extraction techniques is made, and 25 different techniques are identified. Furthermore, this article also explores other design considerations like analytical approaches in the pattern recognition process, operational functionality and system implementation. It is expected that the outcomes of this research may facilitate the researchers and practitioners of the domain during the selection of appropriate feature extraction techniques, machine learning algorithms and other design considerations according to the SHM system requirements.

## 1. Introduction

The construction of cities, villages and related infrastructure are among the unavoidable necessities due to the acceleration of population growth in recent decades. To conduct social, welfare, political and economic activities, a city, nation, or region uses a variety of systems, equipment and services together referred to as infrastructure [1,2,3,4,5]. One of the relatively expensive infrastructures is a bridge that deteriorates with time for a variety of reasons, including creep, corrosion and cyclic loads. However, they can last for hundreds of years with accurate management and maintenance operations by employing an appropriate damage identification process [6,7,8,9,10]. 

The first two levels in a typical damage identification process for bridges are damage identification and damage localization. As the name implies, the damage identification level detects the flaws while the damage localization level determines the position of the flaw in bridges. These two levels fall under the category of diagnosis [11]. The next three levels in a damage identification process are damage classification, damage assessment and life-time prediction. These three levels fall under the category of prognosis. The damage classification level determines the strength type of the flaw while the damage assessment level explores the chances of restricting the flaw. Finally, the life-time prediction evaluates the effect of the flaw on the life cycle of the bridge. 

Historically, the damage identification process has been performed manually to monitor the health of the bridges. However, some major limitations in a manual process are the lack of regular monitoring, inspection-dependent monitoring, the delay in flaw detection and the inability to determine the flaw growth stages [12]. As an alternative to manual monitoring, structural health monitoring systems (SHM) are used. In a typical SHM for bridges, sensors are scattered throughout the structure and the collected data are used to analyze the bridge’s health [13,14,15]. 

The SHM systems were originally constructed with wired sensor networks; however, the great trustworthiness and inexpensive installation and maintenance costs of wireless sensor networks (WSNs) introduce them as a viable option platform [16,17,18,19]. The parameters that can be used to assess the structures using SHM systems are corrosion, cracking, displacement, fatigue, force, settlement, strain, temperature, tilt, vibration, water level and wind [20]. Similarly, the most frequently used sensors in SHM systems are accelerometers and strain sensors. corrosion sensors. linear Voltage Differential Transducers (LVDT) and optical fiber sensors [21,22]. From an implementation point of view, SHM systems can be model-driven or data-driven [23,24,25,26,27,28,29].

The SHM process begins with the “excitation” step that arouses the existing structure looking for a sign of deterioration [30]. The next step is “data acquisition”, which determines how the data may be collected and used. The scaling of inconsistent data, collected from various sensors, is performed in the “data normalization” step. Similarly, the process of identifying and removing errors/duplications in the collected data is also important. It is called the “data cleaning” step. Once the data are acquired, normalized and cleaned, the “data compression” step lowers the feature size to focus on the most statistically significant damage-sensitive aspects. Subsequently, “feature extraction” techniques reduce the feature space by selecting a subset of the original feature such that the number of possible features is minimized. The feature extraction step is followed by the “data fusion” that integrates multiple data sources to produce more accurate, consistent and useful datasets. Finally, the “pattern recognition” step automatically identifies patterns and irregularities in the dataset. 

Among other stages in the SHM system, the feature extraction and pattern recognition stages are critical. The feature extraction stage is a method of transforming the gathered data into a form that can be easily recognized and analyzed by any machine learning algorithm in the pattern recognition stage. The feature extraction can be implemented by signal processing techniques to identify, locate and predict structural health in bridges [31,32,33,34]. On the other hand, several machine learning algorithms have been deployed in the process of pattern recognition [35,36,37,38,39,40,41,42,43,44]. The machine learning algorithms have been deployed in different stages of structures’ life-time including design optimization, performance assessment, maintenance, SHM, damage detection and construction [45,46,47,48,49].

### 1.1. Motivation for a Systematic Literature Review

As stated earlier that the most crucial stages in the process of SHM are the pattern recognition process and the associated feature extraction technique. Various design techniques have been deployed to integrate machine learning algorithms in the pattern recognition process. These techniques employ different types of learning algorithms (supervised, unsupervised and neural network) for different SHM schemes for bridges. Each scheme requires a different analytical approach (classification, regression and clustering). For example, supervised learning algorithms are deployed for classification and regression problems, whereas unsupervised learning algorithms are deployed for clustering problems. Moreover, signal processing techniques are becoming popular in the feature extraction stage of the SHM process, especially with the advancement in the areas of the Internet-of-things (IoT) and big data. Consequently, a systematic study is required to classify and describe the existing design implementation. Furthermore, a systematic study can reveal the need for future studies to overcome the challenges of the existing systems.

### 1.2. Existing Literature Reviews on the Deployment of Machine Learning Algorithms in SHM of Bridges 

Table 1 summarizes the existing literature reviews on the deployment of machine learning algorithms in SHM of bridges [50,51,52,53,54,55,56]. It can be observed from Table 1 that existing reviews describe the current trends in the field with different focus aspects. However, a systematic literature review that presents an in-depth comparative study of different applications of machine learning algorithms in the field of SHM of bridges does not exist. Furthermore, there is a lack of analytical studies that investigate the SHM systems in terms of several design considerations including feature extraction techniques, analytical approaches (classification/regression), operational functionality levels (diagnosis/prognosis) and system implementation techniques (data-driven/model-based). Table 1 overviews the focus and the limitations of the existing literature reviews on the deployment of machine learning algorithms in SHM of bridges. 

### 1.3. Contributions

In order to address the research gap identified in Section 1.2, we conducted an SLR to perform an in-depth analysis of different SHM applications in terms of various algorithms and techniques. Through an SLR approach, the explorations for the answers to the following formulated research questions are the key contributions of this work:

**Research question 1**: What important research has been reported from 2016 to 2023 where machine learning algorithms have been utilized in the pattern recognition process for SHM in bridges?

**Research question 2**: Which of the machine learning algorithms and the analytical approaches are more frequently utilized in the pattern recognition process for SHM in bridges during the 2016–2023 research? 

**Research question 3**: Which of the signal processing techniques are more frequently utilized in the feature extraction process for SHM in bridges during the 2016–2023 research? 

**Research question 4**: Which of the system implementation techniques and operational functionality approaches are more frequently utilized in the process for SHM in bridges during the 2016–2023 research?

### 1.4. Layout of the Research Approach

In order to find responses to the constructed questions (Q1–Q4), the actual layout/framework of the used SLR approach in this article is shown in Figure 1.

Using three scientific databases (i.e., IEEE, Springer, Elsevier), the research studies were carefully selected through some inclusion and exclusion rules. The details of the employed research methodology for the selection of research studies are provided in Section 2. The selected studies were classified into three types: supervised learning algorithms (16 research studies), neural network algorithms (25 research studies) and combined algorithms (4 research studies). To carry out an inclusive examination and synthesis of the selected studies (total = 45), the aforesaid types (supervised, neural network and combined) were further organized according to different mechanisms. Consequently, Section 3 analyzes the selected studies in terms of different design considerations including feature extraction techniques (25 techniques), analytical approaches (classification/regression), operational functionality (diagnosis/prognosis) and system implementation techniques (data-driven/model-based). While a brief overview of the synthesis results of machine learning algorithms is provided in Section 3, an in-depth exploration is provided in Section 4. It includes four algorithms in the supervised category including decision tree (DT), random forest (RF), support vector machines (SVM) and K-nearest neighbors. The neural network category includes artificial neural network (ANN) and convolutional neural network algorithms. Subsequently, responses against the constructed queries are discussed in Section 5. In Section 6, we will discuss the results and the limitations of the research. Finally, the conclusions of this article are provided in Section 7.

## 2. Research Methodology

The systematic literature review process described in [57] was used to carry out this research. It is a proper and replicated process of documenting pertinent details on a precise research area for reviewing and investigating all existing research related to research questions. Consequently, this research incorporates six stages: (1) categories definition, (2) review protocol development, (3) selection and rejection criterion, (4) search process, (5) quality assessment, (6) data extraction and synthesis.

### 2.1. Categories Definition

We defined three categories to organize the selected research studies. This categorization significantly improves the accuracy of the answers to our research questions. The details of the categories are given below.

#### 2.1.1. Supervised Learning Algorithms

Supervised algorithms utilize a labeled dataset. These algorithms are used for classification and regression analytical processes. Consequently, the “Supervised learning algorithms “category includes all those research articles in which one or more than one supervised learning algorithms (random forest, decision tree, support vector machine and K-nearest neighbor) are used for the pattern recognition process of the SHM system for bridges. 

#### 2.1.2. Neural Network Algorithms

Neural network algorithms such as the Artificial Neural Network (ANN) and Convolutional Neural Network (CNN) are designed to perform supervised or unsupervised learning algorithms [49]. Unsupervised learning utilizes unlabeled data for training and is used for clustering and dimensional reduction in analytical processes. The research articles in which one or more than one neural network learning algorithm (ANN and CNN) are used for the pattern recognition process of SHM system for bridges are included in the neural network learning algorithm category. 

#### 2.1.3. Combined Algorithms

The research articles in which a combination of supervised and neural network learning algorithms is used for the pattern recognition process are included in the combined category.

### 2.2. Review Protocol Development

Once the categories are defined, we developed a review protocol for our research on the basis of predefined SLR standards [57]. The developed protocol defines the selection and rejection criterion, search process, quality assessment, data extraction and synthesis of the extracted data. The details of the review protocol are given in subsequent sections.

#### 2.2.1. Selection and Rejection Criterion

We defined a concrete criterion for the selection and rejection of research works. Six parameters were defined to ensure the correctness of the answers to our research questions. The research work is selected based on these parameters as given below:

***Subject relevant:*** Select the research work only if it is relevant to the research context. It must support the answers to our research questions and must be relevant to one of the three predefined categories (Section 2.1). Reject irrelevant research that does not belong to any of the four predefined categories.

***2016–2023:*** Selected research work must be published from 2016 to 2023. Reject all research articles published before 2016 to ensure the inclusion of the latest research works.

***Publisher:*** Selected research work must be published in one of the three renowned scientific databases, i.e., IEEE, SPRINGER and ELSEVIER.

***Crucial effects:*** Selected research work must have crucial positive effects regarding the deployment of machine learning algorithms in the pattern recognition process of SHM for bridges. 

***Results-oriented:*** Selected research work must be results oriented. The proposal and ultimate outcomes of the research must be supported by solid facts and experimentation. Reject the research work if its proposal is verified through a weak validation method.

***Repetition:*** All the research in a particular research context cannot be included. Consequently, reject searches that are identical in the given research context, and only one of them is selected.

#### 2.2.2. Search Process

The selection and rejection criterion, presented in Section 2.2.1, shows that we selected three scientific databases (i.e., IEEE, ELSEVIER and SPRINGER) in order to carry out this SLR. These scientific databases contain high-impact journals and conference proceedings. Furthermore, we also explored related books and technical reports to enhance our knowledge. To accomplish the search process, we used different search terms like SHM, bridges, machine learning, etc. The search terms along with the results for each scientific database are summarized in Table 2.

We used the “*2016–2023*” filter to get research articles published during 2016–2023. The results obtained through the AND operator do not guarantee the relevance of our research context. Therefore, we also used the OR operator to obtain some potential search results required for our research. However, the OR operator provides too many search results to scan all. Consequently, we also used filters to refine the results such as “content type = article”, subject area = Engineering. 

The steps performed during the search process are depicted in Figure 2. It can be observed from Figure 2 that various search terms are specified in three scientific databases. For this review, we analyzed approximately 7832 search results. In the next step, 4861 research articles were discarded by reading their *Title.* We further discarded 1824 research articles by reading their *Abstract*. Subsequently, we performed a general study of 1147 articles by reading different relevant sections of each research. Based on our general study, we discarded 771 articles that do not meet the selection and rejection criterion. We selected the remaining 376 relevant research articles for a detailed study. We performed a detailed study of 376 research articles and discarded 331 research articles. Finally, we selected 45 research articles fully compliant with our pre-defined selection and rejection criteria.

#### 2.2.3. Quality Assessment

We developed quality criteria to understand the important outcomes of selected research studies. The developed criteria also define the credibility of each selected research and its decisive findings: (1)The data appraisal of the research is based on concrete facts and theoretical perspectives without any vague statements.(2)The validation of research has been performed through proper validation methods (case study, etc.).(3)The research provides information about the implementation of the SHM systems.(4)As we intend to investigate the latest machine learning algorithms and trends, the objective is to include the most recent research as much as possible. Therefore, 78% of research articles are from 2020 to 2023. Moreover, 91% of the research articles included are from 2018 to 2023 as shown in Figure 3.

(5)The originality of the research is another important factor. Therefore, we only included articles that are published in at least one of the three renowned and globally accepted scientific databases, i.e., IEEE, SPRINGER and ELSEVIER. In addition, we excluded conference papers from our scope.

#### 2.2.4. Data Extraction and Synthesis

Data extraction and synthesis, as shown in Table 3, were performed to get the answers to our research questions. For data extraction, defined from serial numbers 2 to 4, we extracted important details of each research to ensure its compliance with the selection and rejection criterion. For data synthesis, defined from serial numbers 5 to 9, we performed a detailed analysis of each research. For example, all selected articles were thoroughly studied and analyzed in order to assign them to the corresponding category. Similarly, each selected research was thoroughly analyzed to extract accurate information regarding operational functionality, system implementation, feature extraction, analytical approach and learning algorithm. 

## 3. Results

This section first classifies the selected studies [58,59,60,61,62,63,64,65,66,67,68,69,70,71,72,73,74,75,76,77,78,79,80,81,82,83,84,85,86,87,88,89,90,91,92,93,94,95,96,97,98,99,100,101,102] in terms of pattern recognition algorithms (Section 3.1). Another important aspect in the pattern recognition stage of SHM is the utilization of analytical approaches (Section 3.2). Subsequently, the feature extraction and the associated signal processing algorithms are discussed (Section 3.3). In addition to various steps in the SHM process (such as pattern recognition and feature extraction), there are two additional design parameters that are required to be analyzed. These design parameters are the operational functionality approach (Section 3.4) and the system implementation scheme (Section 3.5). 

### 3.1. Pattern Recognition Algorithms 

Pattern recognition is the last step of the SHM process in which the machine learning algorithms are deployed to automatically identify patterns and regularities in data. In this stage, the decision is taken on the state of the structure. The scope of this SLR is limited to the supervised learning algorithms (DT, RF, SVM, KNN) and the neural network learning algorithms (ANN and CNN) as they are the most used algorithms in the given research context. Table 4 shows the distribution of the selected studies into the predefined categories in Section 2.1. It can be seen from Table 4 that neural network learning algorithms are the most widely used. This is expected because the neural network learning algorithms can be deployed to perform supervised and unsupervised learning processes. An in-depth comparative study of various machine learning algorithms employed in the selected studies for each category is presented in Section 4 of this SLR. 

### 3.2. Utilization of Analytical Approaches

Another important aspect of pattern recognition is the selection of analytical methods applied to recognized patterns. The most analytical classes used in SHM of bridges are classification and regression. Regression techniques are used to predict the output numeric values based on the input characteristics found in the dataset. In contrast, the classification technique produces a class rather than a numeric value. It can be observed from Table 5 that the classification method is frequently employed in the process of SHM for bridges (38 studies out of 45). This is expected because the SHM process is a classification problem from the machine learning point of view to compare damaged and undamaged states of the structure. Only six studies deploy the regression approach where the SHM system is applied to predict the behavior of the bridge under different environmental conditions. For example, DUNWEN et al. [66] deploy the regression approach to predict the hydration heat of mass concrete to apply a temperature control method in advance to reduce the possibility of thermal cracks.

### 3.3. Feature Extraction Techniques Utilization 

As can be seen in Table 6, the most frequent signal processing techniques used in the selected studies in this SLR are FRF, PCA, FFT and GMM methods. Gordan et al. [99] deploy the FRF technique to collect the first four natural frequencies from the experimental dataset to implement data-mining damage identification on a slab-on-girder bridge. Zhang et al. [86] deploy the PCA method to dimensionally reduce the calculated features in the dataset into a small number of features. 

### 3.4. Operational Functionality Investigations 

As mentioned in Section 1, the damage identification process can be classified into five levels: identification, localization, classification, assessment and life-time prediction. The first two levels are used for the diagnosis process of the structural health of the bridge, whereas the latter three are considered prognosis levels to predict the impact of the damage on the structure. 

It can be observed from Table 7 that the diagnosis process is frequently applied in the SHM for bridges (26 studies out of 45). For example, Bing et al. [68] propose a climbing robot to automatically collect impact echo (IE) signals from concrete bridges. These signals are then analyzed to detect the damage. Only 19 selected studies deploy the prognosis process in which the SHM process is deployed in order to classify, assess or predict the damage. For example, Ghiasi et al. [91] develop damage classification systems using vibration-based deep learning approaches. This system can classify different extents of cross-section losses due to corrosion damage. Yanez-Borjas et al. [96] develop a vibration-based methodology in which the autocorrelation of the vibration data is used to detect, locate and assess the corrosion damage in steel truss bridges.

### 3.5. System Implementation Investigations 

From an implementation point of view, SHM systems can be model-driven or data-driven [23]. It can be observed from Table 8 that data-driven is the more commonly used implementation method in the SHM for bridges (27 studies out of 45). 

In the model-based category, the undamaged condition model of the building is created using finite element analysis (FEA) [24]. Model computational complexity is the major limitation of model-based systems [25,26,27,28]. For example, Entezami et al. [61] validate the methodology of bridge condition assessment by numerical concrete beam modeled with 4-node linear 2D elements with reduced integration. The simulation process deployed by the ABAQUS Explicit finite element code. In contrast, data-driven techniques are very practical in handling ambiguity and unexpected cases [29]. Various mechanisms to detect and locate the damage in data-driven implementations are the vibration-based anomaly, vision-based surface crack and sub-surface rebar. For example, Jie et al. [102] develop a bridge SHM system by monitoring different factors, which include strain, temperature, traffic flow and heavy vehicle number. The output of the framework is the health degree of the bridge depending on the classification of different monitoring factors. 

## 4. Machine Learning Algorithms Investigations

Section 3 provides the synthesis/classification results from the selected research studies [58,59,60,61,62,63,64,65,66,67,68,69,70,71,72,73,74,75,76,77,78,79,80,81,82,83,84,85,86,87,88,89,90,91,92,93,94,95,96,97,98,99,100,101,102] in terms of pattern recognition algorithms, utilization of analytical approaches, feature extraction signal processing algorithms, operational functionality approach and the system implementation scheme. However, an in-depth comparative study of the machine learning algorithms employed in the pattern recognition process is one of the core objectives of this SLR. Therefore, this section provides a comprehensive analytical comparison of machine learning algorithms in terms of various performance attributes. A comparison is made for all three categories, as defined in Section 2.1. Firstly, Section 4.1 analyzes the studies in the supervised learning algorithms category, which includes RF, DT, SVM and KNN algorithms. Then, the neural network category is investigated in Section 4.2, which includes ANN and CNN algorithms. Lastly, the studies in the combined category are discussed in Section 4.3. 

### 4.1. Supervised Learning Algorithms 

In the first category (supervised learning algorithms), 16 studies are selected, which cover the following algorithms DT, RF, SVM and KNN. The 16 selected studies for supervised learning algorithms are further categorized as: 2 studies for the RF category, 2 studies for the DT category, 7 studies the for SVM category, 3 studies for the KNN category and 2 studies for the mixed category in which three algorithms are applied in the same study (RF, SVM, KNN) [72,73]. 

#### 4.1.1. Decision Trees (DT)

Decision trees are one of the most widely used algorithms in data mining. The decision tree-based predictive models can be applied to both stratified and regression models [103]. Moreover, Decision trees can be in many forms such as classification and regression trees (CART), Chi-squared automated interaction detection (CHAID), C4.5 and ID3 [104,105]. Table 9 presents an overview of selected research studies in which the DT algorithm has been used. The third column in Table 9 identifies the number of nodes that have been used in the development of the algorithm. The dataset size, that has been used in the training and testing process, is presented in the fourth column. The accuracy of the proposed system in targeted applications is expressed in the last column. 

As can be seen in Table 9, the decision tree algorithm has been deployed in many SHM applications that include damage detection, damage severity prediction and condition assessment of bridges. The accuracy of the algorithm in solving both classification (damage detection tool) and regression (decision-making tool) problems is satisfactory. Furthermore, it can manage numerical and categorical data with minimal data preparation. The main disadvantage of this algorithm is that it is sensitive to specific features in the dataset and therefore small changes to the data result in significant changes to the tree.

For example, Entezami et al. [61] propose a novel SHM methodology to evaluate structural conditions and detect potential damage in civil structures. To examine system reliability, two types of damage-sensitive features were introduced. These are derived from the autoregressive (AR) and principal component analysis (PCA) models. The percentage of the classification error of the DT algorithm according to the AR coefficient and PCA coefficient are 24.37% and 37.5%, respectively. 

#### 4.1.2. Random Forest (RF)

The random forest algorithm is a nonparametric tree-based ensemble technique. It was first proposed by Breiman [106]. Random forests employ a range of understandable decision tree models. Data from several decision tree models can be combined to produce more precise forecasts [107,108]. When dividing a “node” the algorithm looks for the best properties among a random sample of properties rather than the most important attributes. Consequently, a large range of possibilities and a better model can be achieved [109,110,111]. Table 10 presents a review of selected studies in which the RF algorithm was used. The third and the fourth columns in Table 10 identify the number of trees and the minimum leaf size that has been deployed in the development of the algorithm. The dataset size that has been used in the training and testing process is presented in the fifth column. The accuracy of the proposed system in targeted applications is expressed in the last column. 

As shown in Table 10, the random forest algorithm has been deployed in many SHM applications that include damage detection, condition assessment and early damage prediction of bridges. This operation of the RF algorithm is time-consuming and more complex as compared to the decision tree algorithm as it merges individual trees. In addition, the RF algorithm can manage large datasets efficiently, which, on the other hand, increases the number of calculations and the memory overhead. For instance, Dawei et al. [58] propose a non-destructive magnetic flux leakage detection system to precisely identify damage in cable wires. Modeling and simulation approaches were used to design the system. The maximum detection errors of the random forest algorithm in the vertical climbing mode of width and cross-sectional area loss were 0.64 mm and 0.46%, respectively. Whereas the maximum detection errors of the random forest algorithm in the spiral climbing mode of width and cross-sectional area loss were 0.21 mm and 0.1%, respectively. These results show that the spiral climbing mode provides higher classification accuracy. 

#### 4.1.3. Support Vector Machines (SVM)

One of the most efficient supervised machine learning algorithms is the support vector machine, which was proposed by Cortes and Vapnik [112]. One of the crucial factors in the success of the SVM process is the selection of the kernel function in different conditions. This function is responsible for transforming the dataset into the appropriate category [113]. Table 11 presents a review of selected studies in which the SVM algorithm was used. The third column in Table 11 identifies the type of kernel function that has been used in the development of the algorithm. The dataset size, that has been used in the training and testing process, is presented in the fourth column. The accuracy of the proposed system in targeted applications is expressed in the last column. 

As shown in Table 11, the SVM algorithm has been deployed in many SHM applications that include damage identification, localization, condition assessment and damage prediction of bridges. This algorithm works well on small datasets. The SVM algorithm is the most deployed supervised algorithm as it exhibits a relatively high accuracy in solving both classification (damage detection tool) and regression (early prediction tool) problems. The main disadvantage of the SVM algorithm is that the training time increases significantly with the dataset size. Yifu et al. [64] propose a data-driven SHM system based on Optimized AdaBoost-Linear SVM. The main objective of the system is to identify damage by means of vibration signals received from a vehicle passing over the bridge. In the implementation stage, the AdaBoost-SVM methodology increases the accuracy of the results by 5% to 16.7% compared to other algorithms such as SVM and RF. 

#### 4.1.4. K-Nearest Neighbors (KNN)

The K-Nearest Neighbors method (KNN) algorithm is a supervised learning algorithm that is considered one of the simplest algorithms in terms of application complexity. The most important parameter that affects the operation of the KNN algorithms is the number of neighbors, which is highly influenced by the noise in the dataset. Furthermore, the size of the dataset needed for training the algorithm is directly proportional to the dimension of the feature under investigation, which, in turn, increases the computational cost of the system [114,115]. Table 12 presents a review of selected studies in which the KNN algorithm was used. The third column in Table 12 identifies the number of neighbors that have been used in the development of the algorithm. The dataset size that has been used in the training and testing process is presented in the fourth column. The accuracy of the proposed system in targeted applications is expressed in the last column.

As can be seen in Table 12, the KNN algorithm has been deployed in many SHM applications that include damage identification, localization and condition assessment. This algorithm is used only to solve the classification problems in the selected studies with acceptable accuracy. The KNN algorithm does not work well with imbalanced data and the prediction process runs quite slowly when working with large datasets. For example, Sarmadi et al. [69] propose a novel anomaly detection system for SHM under variable environmental conditions. The system deploys a mechanism called adaptive Mahalanobis-squared distance and one-class KNN (AMSD-kNN). The main objective of the system is to identify the appropriate nearest neighbors for training and testing datasets in order to eliminate the environmental effect of the variable environmental conditions. In the implementation stage, the total error is 0.25% for 90% of the learning sample. 

Svendsen et al. [72] introduce a data-based SHM approach for damage identification in steel bridges. Two groups of sensors have been utilized for acquiring both the local and global responses of the considered bridge. Four supervised machine learning algorithms including the k-nearest neighbors (kNN), the support vector machine (SVM), the random forests (RF) and the Gaussian naïve Bayes (NB) algorithms were applied to assess the capabilities of these algorithms to identify and classify structural damage. The system accuracy was satisfactory for SHM applications with a total error of type 1 and type 2 error of 19.7%.

Wang et al. [73] present an effective label ranking method for bridge condition assessment (LR-BCA). The mechanism of the system operation investigates the natural order relationship between bridge condition ratings. Furthermore, the feature extraction process is accomplished by the means of a heuristic data cleaning (HDC) technique. The HDC method was applied for cleaning the bridge condition dataset by recognizing all the label conflict samples, then iteratively filtering out the noise. Then, three supervised machine learning algorithms including RF, SVM and KNN were deployed to evaluate the effectiveness of HDC in LR tasks. The proposed LR-BCA approach attains an accuracy of 99% in predicting different bridge conditions ratings.

### 4.2. Neural Network Learning Algorithms 

In the second category (neural network algorithms), 23 studies were selected, which cover the following algorithms ANN and CNN algorithms. The 23 selected studies for NN learning algorithms are further categorized as: 12 studies for the ANN category and 13 studies for the CNN category.

#### 4.2.1. Artificial Neural Network (ANN)

The operation of the artificial neural network (ANN) depends on isolating the input into several levels of abstraction. This network can be trained using datasets to recognize patterns in images or sounds. The optimal behavior of the network can be achieved by the methodology of connections between different components in the network and the weight of those components. An automated process is normally utilized to modify the component’s weight during the training process [116,117]. Table 13 presents a review of selected studies in which the ANN algorithm was used. Columns 3, 4, 5 and 6 in Table 13 identify the number of neurons, the number of hidden layers, the training algorithm and the activation function that has been used in the development of the algorithm. The dataset size that has been used in the training and testing process is presented in the seventh column. The accuracy of the proposed system in targeted applications is expressed in the last column.

As can be seen in Table 13, the ANN algorithm has been deployed in many SHM applications that include damage identification, localization, condition assessment and damage prediction. The ANN algorithm is one of the most deployed neural network algorithms as it exhibits a relatively high accuracy in solving both classification (damage detection tool) and regression (early prediction tool) problems. The ANN algorithm requires a large dataset during training which in turn increases the computational cost of the proposed system. For instance, Hooman et al. [78] propose an ANN-based two-level damage identification approach to damage localization and severity estimation in steel girder bridges. The reliability of the system was examined by the means of a finite element (FE) model of the I-40 Bridge. In the validation stage of the system, the accuracy of the results was sufficiently high with a maximum modal strain energy-based damage index error of 1.2%.

#### 4.2.2. Convolutional Neural Network (CNN)

The convolutional neural network (CNN) is a feed-forward neural network, which is the most commonly used deep learning approach in the area of image and object classification. CNN consists of many layers including a convolutional layer, pooling layer, ReLU correction layer and fully connected layers. It is widely used for 2D image classification [118]. The XAI (eXplainable Artificial Intelligence) techniques can be used to interpret the obtained results from the CNN algorithm in a form that is humanly explainable and directly implementable in new tools for bridge inspections. This facilitates the observation of the activation zones and nearly perfectly highlights the type of specific defect in a given image [119]. Table 14 presents a review of selected studies in which the CNN algorithm was used. Columns 3, 4 and 5 in Table 14 identify the convolution layers size, pooling layers size and the activation function that has been used in the development of the algorithm. The dataset size that has been used in the training and testing process is presented in the sixth column. The accuracy of the proposed system in targeted applications is expressed in the last column.

As shown in Table 14, the CNN algorithm has been deployed in many SHM applications that include crack detection, anomaly detection, damage localization and crowd estimation. The CNN algorithm is one of the most deployed deep learning algorithms as it is computationally efficient with large datasets and exhibits a relatively high accuracy in solving both classification (crack detection tool) and regression (crowd estimation tool) problems. For example, Dinh et al. [98] develop a based automated rebar localization and identification approach. The proposed system integrates conventional image processing methods and convolutional neural networks (CNN). In the validation stage, the overall damage identification accuracy of the system was found to be 99.60% ± 0.85%. 

### 4.3. Combined Category

In the third category (combined algorithms), four studies have been selected. Two of them have combined CNN and SVM algorithms [101,102]. Similarly, the work in [99] combines ANN, SVM and DT algorithms [99]. Finally, the work in [100] combines CNN and KNN algorithms [100]. The review of these studies is covered in Section 4.1 and Section 4.2. 

The proposed systems in the combined category utilize different machine learning algorithms for different purposes. For example, the ANN, SVM and DT algorithms have been used in [99] as a tool of comparison to validate the performance of the proposed model, which is the Cross Industry Standard Process for Data Mining (CRISP-DM) model for damage severity assessment. On the other hand, the KNN algorithm has been used as a feature extraction technique in [100] to facilitate the pattern recognition process executed by the CNN algorithm. Similarly, the CNN algorithm has been used as a feature extraction technique in [102] to facilitate the pattern recognition process executed by the SVM algorithm. The proposed system in [101] deployed both SVM and CNN algorithms to solve a classification problem (crowd attribute classification for motion speed and load designation) and a regression problem (load estimation). This integration of machine learning algorithms increases the efficiency of the proposed SHM systems in damage detection. 

Parisi et al. [100] propose a damage identification approach in steel truss railway bridges by deploying machine learning classification algorithms. The proposed method eliminates the need for feature extraction techniques in the analysis of the signal from the strain sensor. The proposed system integrates the K-nearest neighbors and the convolutional neural network algorithms for feature extraction and classification purposes, respectively. The system accuracy of damage detection is 93%.

Samir et al. [101] deploy the latest module for simultaneous crowd and structural monitoring, which implements the integration of sensing technologies (Fiber Bragg Grating (FBG) and Fiber Optic Sensors (FOSs)) with wearable sensing devices incorporating Inertial Measurement Units (IMUs). The proposed system integrates CNN and SVM for the pattern recognition stage. The accuracy of the system is sufficiently high with peak testing accuracy for single-class motion speed classification at 98%, multi-class motion speed and load characterization classification at 91%, and percentage error for load estimation regression reaching a minimum of 9%.

## 5. Answer Research Questions

**Research question 1**: What important research has been reported from 2016 to 2023 where machine learning algorithms have been utilized in the pattern recognition process for SHM in bridges?

**Answer**: 45 important research studies published from 2016 to 2023 have been recognized as per the selection and rejection criterion (Section 2.2.1). These researches are classified into three corresponding categories. The details are as follows:Sixteen research studies have been recognized in the supervised learning algorithms category (Section 4.1).Twenty-five research studies have been recognized in the neural network learning algorithms category (Section 4.2).Four research studies have been recognized in the supervised learning algorithms category (Section 4.3).

**Research question 2**: Which of the machine learning algorithms and the analytical approaches are more frequently utilized in the pattern recognition process for SHM in bridges during the 2016–2023 research?

**Answer:** On the basis of this SLR, the accuracy of the neural network learning algorithms including the ANN and the CNN techniques in anomaly and crack detection is above 80%. Furthermore, one of the most important features of neural networks is that they can be deployed in supervised (classification, regression) and unsupervised learning (clustering) contexts. Consequently, they are the most used algorithms in the pattern recognition process for SHM in bridges. Further details are available in Table 13 and Table 14. Furthermore, from the selected studies, it can be seen that the classification method is the most commonly used technique applied in the process of SHM for bridges (38 studies out of 45) as shown in Table 5. This is expected because the SHM process is a classification problem from the machine learning point of view to compare damaged and undamaged states of the structure. Only six studies deploy the regression approach in which the SMM system is applied in order to predict the behavior of the bridge under different environmental conditions.

**Research question 3**: Which of the signal processing techniques are more frequently utilized in the feature extraction process for SHM in bridges during the 2016–2023 research?

**Answer**: on the basis of this SLR, the signal processing techniques most deployed in the feature extraction process for SHM in bridges are FRF, PCA and FFT methods. These methods enhance the system’s efficiency by reducing the processing time required for damage recognition. Further details are available in Table 6. There are other approaches that have been deployed in the bridges SHM applications such as wavelet transform (discrete and continuous methods), Kalman filter-based techniques and the Autocorrelation method.

**Research question 4**: Which of the system implementation techniques and operational functionality approaches are more frequently utilized in the process of SHM in bridges during the 2016–2023 research?

**Answer**: From the selected studies, it can be seen that the diagnosis process is the most technique used in the SHM for bridges (26 studies out of 45) as shown in Table 7. Only 19 selected studies deploy the prognosis process in which the SHM process is deployed in order to classify, assess or predict the damage in the bridge structure. In addition, from the selected studies, it can be seen that data-driven is the most implemented style applied in the SHM for bridges (27 studies out of 45) as shown in Table 8. These studies deploy different mechanisms to detect and locate the damage such as vibration-based anomaly detection, vision-based surface crack detection and sub-surface rebar detection. The model-based implementation was applied in 18 selected studies.

## 6. Discussion and Limitations 

**Discussion on learning algorithms for pattern recognition process:** In this SLR, we focused on the applications of machine learning algorithms in the pattern recognition process of SHM systems for bridges. In addition, we restricted our focus to the supervised and neural network learning algorithms as unsupervised learning algorithms are less likely to be used and their applications are limited to clustering problems only. The supervised learning algorithms, discussed in this SLR, include RF, DT, SVM and KNN. It is important to note that there are some other supervised learning algorithms such as Bayesian [120,121,122,123] and ensemble methods [124] that have been used in SHM systems. However, they have not been frequently deployed in the pattern recognition process of SHM systems for bridges. Consequently, the scope of this SLR focuses only on the most widely used learning algorithms including RF, DT, SVM and KNN. In addition, the neural network algorithms (ANN and CNN) are presented in this SLR. From the results of this SLR, it can be argued that the ANN and CNN (55% of the selected studies) are the most widely deployed algorithms due to their wide range of functionality in different SHM applications such as crack detection and anomaly detection. Furthermore, the accuracy of neural network algorithms in damage detection is considerably high as compared to other algorithms as shown in Table 13 and Table 14.

**Discussion on analytical approaches:** The supervised machine learning algorithms are capable of two main analytical approaches: classification and regression. The classification approach can be used in SHM systems to cover the first four levels of damage identification including identification, localization, classification and assessment. On the other hand, the regression approach is limited to prediction studies only. It can be argued that the classification approach is the most analytical approach implemented in the SHM system for bridges as can be seen in Table 5, which is expected because the SHM process is a classification problem from the machine learning point of view to compare between damaged and undamaged states of the structure. In the selected studies for this SLR, only 15.5% of the selected studies deploy regression for prediction operations [59,60,65,66,76,82,101]. For example, Xiao-Wei et al. [59] propose a data-driven approach to predict the vibration amplitudes of girders and towers for early warning SHM. The regression module of the RF learning algorithm was used in the prediction process. A cable-stayed bridge with a main span of 1088 m is taken as the case study. The prediction accuracy of the proposed system according to the validation process is 92.5 %.

**Discussion on feature extraction techniques:** The feature extraction process is an important stage in the SHM procedure. Here, the datasets from different sensors are processed to extract the relevant feature from the design perspective. In other words, feature extraction is implemented by signal processing techniques such as wavelet-based techniques, frequency–response functions (FRF), soft computing-based techniques and Kalman filter-based techniques. As can be seen in Table 6, 38% of the selected studies deploy FRF, PCA, FFT and GMM methods for the feature extraction process. Other techniques identified in this SLR are particle swarm optimization, cuckoo search and heuristic data cleaning. Tran et al. [74] develop a novel approach to detect structural deterioration. The system integrates the ANN algorithm and the evolutionary algorithm cuckoo search (CS) method. In the validation process, two numerical models were used to assess the reliability of the system (a steel beam calibrated using experimental measurements and a large-scale truss bridge). The proposed technique exhibited high accuracy for damage identification (location and severity) with a learning coefficient (R) higher than 0.99 in all test cases. In addition, the system has significantly reduced the computational time. 

**Discussion on operational functionality:** The damage identification process is classified into five levels including identification, localization, classification., assessment and lifetime prediction. The first two levels are used for the diagnosis process of the structural health of the bridge, whereas the latter three are considered prognosis levels to predict the impact of the damage on the life-time of the structure. As can be seen in Table 7, 58% of the selected studies for this SLR deploy diagnosis operations, while 42% of the selected studies perform prognosis methods for SHM for bridges. Thanh cuong et al. [63] develop an SHM system that integrates the particle swarm optimization method and the Support Vector Machine algorithm (PSO-SVM) for structural damage identification, location and severity. The damage location classification accuracy is enhanced by the effective searching capability of PSO, which eliminates the redundant input parameters. The proposed system presents an impressive classification accuracy of 0.0461 and 0.957 for root mean square error (RMSE) and correlation coefficient (R^2^), respectively. 

**Discussion on system implementation:** The physical implementation of the SHM system for bridges can be classified into two main categories: data-driven and model-based approaches. The operation of model-based systems has many computational limitations, therefore, a data-driven system that incorporates different sensing capabilities has recently been deployed more often in SHM systems for bridges. As can be seen in Table 8, 60% of the selected studies for this SLR deployed the data-driven approach, whereas the model-based approach was used in 40% only. For instance, Rageh et al. [82] propose an SHM system for automated damage identification (location and intensity) by means of a continuous stream of data. The system deployed a number of strain sensors on the structure of the steel truss bridge. The focus of the study was to detect the stringer–floor beam connection deterioration. An integration of the ANN algorithm and proper orthogonal modes (POMs) approach is presented in the study. The system showed improved accuracy in damage detection for damage intensities higher than 40%. 

**Limitations of research:** Even though, we have strictly followed the guidelines of SLR presented by Kitchenham [57] and completely observed the developed review protocol, there are some minor limitations:We utilized the relevant search terms and systematically scanned the search outcomes. Nevertheless, a few search terms obtained thousands of outcomes that we could not scan comprehensively. In addition, we excluded several studies on the basis of their titles in accordance with the search process. Therefore, there is a possibility that the scope of the article is not appropriately clear in the title. Subsequently, we do not claim the comprehensiveness of our research in this SLR.We used three prestigious scientific databases, i.e., IEEE, ELSEIVER and SPRINGER, which contain a huge number of journal and conference publications. Nevertheless, there exist other databases that provide a lot of publications. Consequently, there is a fair possibility that we could have excluded recent research from other databases. However, we firmly believe that the final results of this SLR are not considerably affected because high-quality recent research is available in the selected scientific databases.

## 7. Conclusions

This research presents the applications of machine learning algorithms used for the pattern recognition process for SHM in bridges. To accomplish this objective, SLR has been performed to recognize 45 research articles. On the basis of different machine learning techniques, selected research studies are classified into three different categories, and six learning algorithms are discussed. Consequently, a comprehensive analysis is performed on the identified algorithms by considering various important parameters according to the type of the algorithm. Furthermore, the selected studies have been analyzed in terms of different design considerations including the feature selection approaches, the operational functionality, the analytical approaches and system implementation. Thus, the latest applications of machine learning algorithms in the process of pattern recognition for SHM in bridges are presented and analyzed under this SLR, which is rarely available to the best of our knowledge. This will facilitate the selection of the appropriate machine learning algorithm according to the SHM system requirements.

On the basis of this SLR, it can be claimed that the classification method and neural network learning algorithms including the ANN and CNN are the most used techniques and algorithms in the pattern recognition process for SHM in bridges in recent studies. Furthermore, the most signal processing techniques deployed in the feature extraction process for SHM in bridges are the FRF, PCA and FFT methods. In addition, the diagnostic techniques and the data-driven approach are the most widely used operational functionality and system implementation techniques for SHM in bridges, respectively.

The learning algorithms and signal processing techniques considered in this SLR exhibit limitations including the computational complexity, memory requirement and time consumption in the pattern recognition and feature extraction processes, respectively. Accordingly, further investigations of novel combined algorithms and techniques to overcome these limitations should be considered in future research studies. Furthermore, most of the recent studies in the field focus only on the investigations of the damage diagnosis levels including damage identification and localization. Consequently, future research studies should further investigate the implementation of prognosis levels to classify, assess and predict the lifetime of the bridge. This will increase the efficiency of the SHM system in damage detection and increase the lifetime of the bridge.

## Figures and Tables

**Figure 1 sensors-23-04230-f001:**
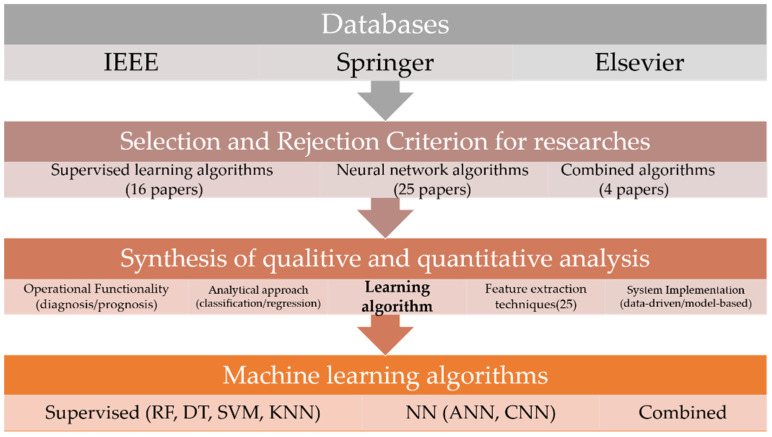
Overview of research.

**Figure 2 sensors-23-04230-f002:**
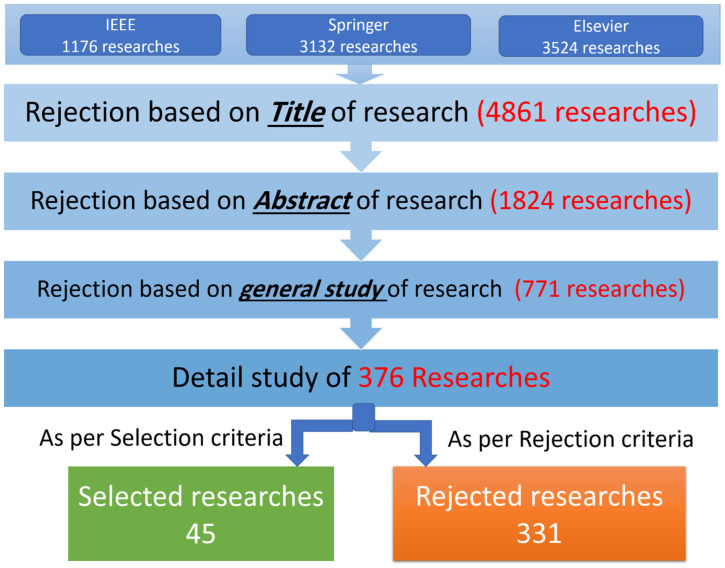
The search process for the conduction of systematic literature review.

**Figure 3 sensors-23-04230-f003:**
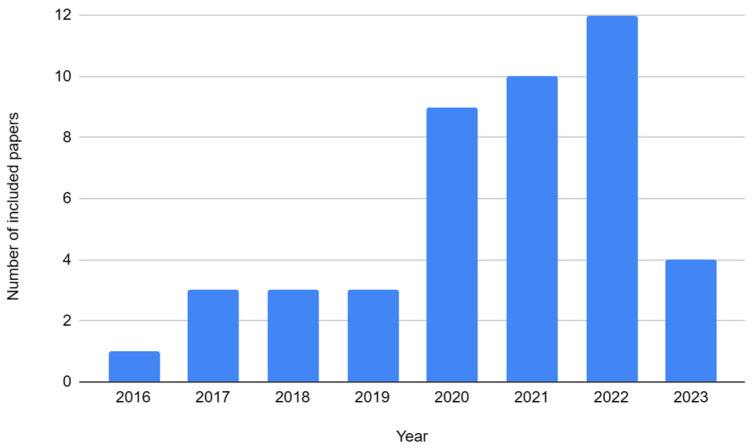
Number of selected research articles per year.

**Table 1 sensors-23-04230-t001:** The existing literature reviews on the deployment of machine learning algorithms in SHM of bridges.

Ref.	Year	Focus	Limitations
[50]	2022	Investigates the use of artificial intelligence to enhance the operation of data-driven SHM systems for bridges.	It is not systematic Limited to data-driven Feature extraction step was not explored Did not investigate the analytical approaches
[51]	2022	Investigates the feature extraction and pattern recognition processes of SHM systems for building and bridges.	It is not systematic It is not focused on bridges Did not investigate the analytical approaches
[52]	2022	Targets the study of vibration-based systems and machine learning techniques in bridges.	It is not systematic Limited to vibration-based Feature extraction step was not explored Analytical approaches were not explored
[53]	2022	Explores the latest trends and limitations of the use of deep learning algorithms in SHM for bridges.	It is not systematic Limited to deep learning algorithms Limited to vibration-based and vision-based Limited to damage detection (diagnosis) Feature extraction step was not explored Did not investigate the analytical approaches
[54]	2021	Evaluates machine learning algorithms in SHM systems for reinforced concrete bridges. It explores the deployment of machine learning algorithms in structural design, construction quality management, bridge engineering and the inspection process.	It is not systematic Limited to reinforced concrete bridges Did not investigate the feature extraction techniques Did not investigate the analytical approaches
[55]	2020	Investigates the latest progress in platforms, sensors, and algorithms in the area of autonomous robotics for the SHM for bridges.	It is not systematic Limited to surface cracks and sub-surface-level rebars detection. Feature extraction step was not explored Did not investigate the analytical approaches
[56]	2023	Explores the applications of artificial intelligence and trending technologies such as drone technology and 3D printers in SHM systems for bridges.	It is not systematic Focus on descriptive analysis (not application oriented) Feature extraction step was not explored Did not investigate the analytical approaches

**Table 2 sensors-23-04230-t002:** Details of search terms with operators and search results.

Serial No.	Search Terms	Operator	No. of Search Results		
			IEEE	Springer	Elsevier
1	‘Bridges’ ‘SHM’ ‘machine learning’	AND	3	141	413
		OR	34,759	31,372	9312
2	‘Bridges’ ‘SHM’ ‘pattern recognition’	AND	0	105	234
		OR	13,415	17,136	3638
3	‘Bridges’ ‘ANN’	AND	1	1249	193
		OR	10,904	30,545	3939
4	‘Bridges’ ‘CNN’	AND	2	594	198
		OR	16,084	25,401	5258
5	‘Bridges’ ‘Random forest’	AND	3	526	158
		OR	9475	5725	3566
6	‘Bridges’ ‘Decision tree’	AND	1	1063	238
		OR	9027	11,159	5351
7	‘Bridges’ ‘K-nearest neighbor’	AND	0	287	121
		OR	8516	6233	5272
8	‘Bridges’ ‘Support vector machine’	AND	5	2695	287
		OR	14,815	28,364	5509

**Table 3 sensors-23-04230-t003:** Details of data extraction and synthesis.

S. No.	Description	Details
1	bibliographic information	Title, author, publication year, publisher details
**Extraction of data**		
2	Overview	The basic proposal and objective of the selected research
3	Results	results acquired from the selected research
4	Validation	The validation method used to validate its system
**Synthesis of data**		
5	Learning algorithm	Relevance to one of the predefined categories
6	Analytical approach	The type of analytical approach (Classification or regression)
7	Feature extraction techniques	Signal processing techniques used to perform feature extraction
8	Operational functionality	The purpose of the SHM system (damage diagnosis, damage prognosis)
9	System implementation	The type of the system (model-based or data-driven)

**Table 4 sensors-23-04230-t004:** Learning algorithms statistics.

S. No.	Category	Number of Researches	Reference
1	Supervised learning algorithms	16	[58,59,60,61,62,63,64,65,66,67,68,69,70,71,72,73]
2	Neural network learning algorithms	25	[74,75,76,77,78,79,80,81,82,83,84,85,86,87,88,89,90,91,92,93,94,95,96,97,98]
3	Combined algorithms	4	[99,100,101,102]

**Table 5 sensors-23-04230-t005:** Statistics for employed analytical approaches in the selected research studies.

S. No.	Category	Number of Researches	References
1	Classification	38	[58,61,62,63,64,67,68,69,70,71,72,73,74,75,77,78,79,80,81,83,84,85,86,87,88,89,90,91,92,93,94,95,96,97,98,99,100,102]
2	Regression	6	[59,60,65,66,76,82]
3	Classification and Regression	1	[101]

**Table 6 sensors-23-04230-t006:** Feature extraction techniques utilization statistics.

Serial No.	Category	Number of Researches	Reference
1	FRF (frequency response function)	5	[75,77,78,82,99]
2	PCA (principle component analysis)	5	[61,64,77,83,86]
3	median filtering	1	[58]
4	wavelet transform methods	1	[58]
5	AR (autoregressive model)	2	[61,72]
6	PSO (Particle swarm optimization)	2	[62,63]
7	Gaussian Cumulative Density Function (CDF)	1	[67]
8	FFT (Fast Fourier transform)	4	[68,80,91,101]
9	DWT (discrete wavelet transform)	1	[68]
10	Short-time Fourier transform (STFT)	1	[70]
11	Continuous Wavelet Transform (CWT)	1	[71]
12	Bayesian Modal Identification method	1	[71]
13	heuristic data cleaning (HDC)	1	[73]
14	Cuckoo search (CS)	1	[74]
15	kernel principal component analysis (KPCA)	1	[83]
16	Gaussian mixture model (GMM)	3	[59,65,83]
17	EKF (Extended Kalman filter)	1	[84]
18	Gaussian process (GP)	1	[85]
19	mean reduction process	1	[92]
20	Autocorrelation method (ACM)	2	[93,96]
21	Fourier transform	1	[93]
22	Normalized cross-correlation	1	[98]
23	Dynamical Time Warping (DTW)	1	[100]
24	Constrained observability method (COM) and	1	[60]
25	Generalized extreme value (GEV) distribution	1	[69]

**Table 7 sensors-23-04230-t007:** Operational functionality statistics in terms of damage diagnosis and prognosis.

S. No.	Category	Number of Researches	Reference
1	Damage diagnosis (identification, localization)	26	[58,64,67,68,69,72,76,77,80,81,82,83,84,85,86,87,88,89,90,93,94,95,97,98,100,102]
2	Damage prognosis (classification, assessment, life -time prediction)	19	[59,60,61,62,63,65,66,70,71,73,74,75,78,79,91,92,96,99,101]

**Table 8 sensors-23-04230-t008:** System implementation statistics in terms of data-driven and model-based methods.

S. No.	Category	Number of Researches	Reference
1	Data-driven	27	[58,59,62,64,66,68,72,73,75,77,79,80,81,82,83,86,87,90,92,93,94,96,97,98,99,101,102]
2	Model-based	18	[60,61,63,65,67,69,70,71,74,76,78,84,85,88,89,91,95,100]

**Table 9 sensors-23-04230-t009:** Review of DT algorithm research studies in the area of SHM of bridges.

Ref.	Bridge Type	No. of Nodes	Dataset Size	Purpose	Accuracy
[60]	pre-stressed concrete bridge	9	1000 samples	Investigates the role of the SHM strategy and the SSI (Structural System Identification) analysis based on the constrained observability method (COM) and decision trees (DT) in reducing estimation error	Mean of error index: 18.39%
[61]	cable-stayed bridges	Not specified	408 samples	A condition assessment and damage detection system for civil structures	Classification error: AR coefficient: 24.37% PCA coefficient: 37.5%
[99]	slab-on-girder bridge	Not specified	25 damage severities	Data-mining–based damage identification approach to predict damage severity by the implementation of CRISP-DM model.	MAE (Mean Absolute Error) 4.706 for training/7.2 for testing

**Table 10 sensors-23-04230-t010:** Review of RF algorithm research studies in the area of SHM for bridges.

Ref.	Bridge Type	No. of Trees	Min. Leaf Size	Dataset Size	Purpose	Accuracy
[72]	Steel bridges	(200, 100)	(4, 4)	120 tests.	Data-based SHM approach for damage detection	Total error: 19.7%
[73]	highway bridge	150	Not specified	7870 examples	An effective label ranking approach for bridge condition assessment	99%
[58]	cable-stayed bridge	10	-	22 defectives samples	Nondestructive magnetic flux leakage detection system for quantitatively identifying wire defects	In the vertical climbing mode, the maximum detection errors in the width and cross-sectional area loss are 0.64 mm and 0.46%, respectively, while the values are 0.21 mm and 0.1% in the spiral climbing mode.
[59]	cable-stayed bridge	200	5	1410 samples	A data-driven approach to predict the vibration amplitudes of girders and towers for early warning	92.5%

**Table 11 sensors-23-04230-t011:** Review of SVM algorithm studies in the area of SHM of bridges.

Ref.	Bridge Type	Type of Kernel	Dataset Size	Purpose	Accuracy
[62]	cable-stayed bridges	radial basis	160 surfaces defect images	Classification of cable surface defects	classification accuracy: 96.25%.
[63]	Truss bridge	GRBF (Gaussian radial basis function)	1000 samples	Damage identification, localization and assessment.	Classification accuracy RMSE = 0.0461 R^2^ = 0.957
[72]	Steel bridges	Linear	120 tests.	Data-based SHM approach for damage detection	Total error: 19.7%
[73]	highway bridge	radial basis function (RBF)	7870 examples	An effective label ranking approach for bridge condition assessment	99%
[64]	Steel bridge	linear	8400 signals	A data-driven approach to indicate the bridge damage using only raw vibration signals received from a vehicle passing over the bridge.	improves result accuracy by 5% to 16.7%.
[101]	pedestrian bridge	Gaussian, quadratic and cubic polynomial	488 observations.	Estimation of crowd flow and load on pedestrian bridges based on a novel combination of sensing technologies that include the employment of structurally mounted Fiber Bragg Grating (FBG) Fiber Optic Sensors (FOSs) in conjunction with individually held wearable sensing devices incorporating Inertial Measurement Units (IMUs).	peak testing accuracy for single class motion speed classification at 98%, multi-class motion speed and load characterization classification at 91%, percentage error for load estimation regression reaching a minimum of 9%.
[65]	cable-stayed bridge	Gaussian kernel function	300 samples	Proposes probabilistic fatigue damage assessment under coupled dynamic loads	It is observed that the SVR response surface is close to all the samples with a maximum absolute difference of less than 1.3%
[66]	cable-stayed bridge	the radial basis kernel function (RBF)	102 measurement points	Prediction of Hydration Heat of Mass Concrete pile caps	The squared correlation coefficient (*R*^2^) of the training and testing sets can reach above 0.99 and 0.98, respectively
[102]	Steel bridge	linear, polynomial, radial basis function (RBF) and sigmoid	151,238 instances	Proposes an end-to-end framework to evaluate the health of bridges by exploring objective features and correlations of multiple monitoring factors.	F1 score, precision and recall, specificity (SPC), G-Mean (G-M) and F-Measure (F-M) above 0.93 for all above matrices
[99]	slab-on-girder bridge	RBF, Polynomial, Sigmoid and Linear	25 damage severities	Data-mining–based damage identification approach to predict damage severity by the implementation of CRISP-DM model.	MAE (Mean Absolute Error) 5.056 for training/4.925 for testing
[67]	Steel bridge	RBF	2400 samples	Detection of distortion-induced fatigue cracking based on the data provided by self-powered wireless sensors	Accuracy: 85%
[68]	Concrete bridge	Linear	800 samples	Damage detection by automated impact echo signal collection using robots.	Accuracy: 99.2%

**Table 12 sensors-23-04230-t012:** Review of KNN algorithm studies in the area of SHM for bridges.

Ref.	Bridge Type	No. of Neighbors	Dataset Size	Purpose	Accuracy
[69]	Wooden Bridge and concrete box girder bridge	595	5652 samples	Anomaly detection method for SHM under environmental effects	For 90% learning sample, the total error is 0.25%
[70]	Long span bridge	-	Four new testing samples, each consisting of 1600 dynamic transient runs,	Locating and quantifying stiffness loss in a bridge from the forced vibration due to a truck crossing at low speed	94%
[71]	Steel railway bridge	1	12 damage cases for training and 6 damage cases for test	Proposes a novel damage detection approach for the classification of various extents and degrees of cross-section losses due to damages like corrosion.	100%
[100]	steel truss railway bridges	Not specified	500 Epochs	A method of locating damage in bridges through classification tools, enabling automatic analysis of raw strain sensors signals without any pre-processing or preliminary feature extraction.	93%
[73]	highway bridge	7	7870 examples with 422 attributes and 3 labels	An effective label ranking approach for bridge condition assessment	99%
[72]	steel bridges	(1, 5)	120 tests	Data-based SHM approach for damage detection	Total error: 19.7%

**Table 13 sensors-23-04230-t013:** Review of ANN algorithm studies in the area of SHM of bridges.

Ref.	Structure	No. of Neurons	No. of Hidden Layers	Training Algorithm	Activation Function	Dataset Size	Purpose	Accuracy
[74]	Steel truss bridge	Not specified	1	Levenberg-Marquardt algorithm based on backpropagation (LMBP)	sigmoid	1500 data samples	Identify damage location and severity.	learning coefficient (R) in all cases is higher than 0.99
[75]	slab-on-girder bridge	15 (input), 3 (output)	1	Imperial competitive algorithm (ICA)	Not specified	Not specified	Prediction of the severity and location of damage using Data-mining-based damage detection methodology	efficiency coefficient (R^2^) 0.998
[99]	slab-on-girder bridge	15 (input), 1 (output)	1	ICA	Not specified	25 damage severities	Data-mining—based damage identification approach to predict damage severity by the implementation of CRISP-DM model.	MAE (Mean Absolute Error) 1.355 for training/2.097 for testing
[76]	cable-stayed bridge	22 (input), 44 (hidden)1 (output)	1	LMBP	Not specified	32,000	Anomaly detection under complex loading conditions	R^2^ 0.9905 MSE 5.6167 MAE 1.8159
[77]	steel truss bridge	20	1	LMBP	tangent sigmoid	1200 samples	Non-probabilistic method to consider uncertainties in frequency response function for vibration-based damage detection	The possibility of damage existence (PoDE) is above 95%
[78]	steel girder bridges	3 (input), 10 (hidden)3 (output)	1	LMBP	Not specified	48 samples for single damage quantification and 64 samples for multiple damage quantification	A two-stage damage identification technique to locate and estimate damage.	modal strain energy-based damage index (maximum error of 1.2%)
[79]	steel bridges	25	2	LMBP	Sigmoid function	1300 samples	Data-driven fatigue assessment system of welded structural components	Not specified
[80]	slab-on-girder bridge	3 (input), 40 (hidden)1 (output)	2	LMBP	hyperbolic tangent	900 passages	A data-driven approach for drive-by damage detection in bridges considering the influence of temperature change	The algorithm is shown to be capable of detecting damage at midspan and quarter-span even at damage levels as low as 5% with 3% and 5% measurement noise
[81]	steel arch bridge	40	1	Bayesian regularization backpropagation	ReLU	514 measurements	A model-free damage detection method	Not specified
[82]	Steel truss bridge	100	1	Bayesian regularization and early cessation	Nonlinear	2800 damage scenarios	Automated damage detection using a continuous stream of SHM data.	The accuracy is improved for damage intensity (DI) higher than 40%.
[83]	concrete box girder bridge	50	2	Not specified	ReLU	600 samples	Anomaly detection in a bridge from vibrational measurements using the minimum amount of data.	accuracy greater than 94%
[84]	steel truss bridge	6	1	BP	Tangent	6480 samples	Damage detection method under temperature changes	MAE (Mean absolute error) = 0.0572
[85]	single track railway bridge	49 (input), 30 (hidden)1 (output)	1	Not specified	Not specified	300 train passage	A model-free damage detection approach	Root Mean Squared Error (RMSE) = 0.2

**Table 14 sensors-23-04230-t014:** Review of CNN algorithm studies in the area of SHM of bridges.

Ref.	Structure	Convolution Layers Size	Pooling Layers Size	Activation Function	Dataset Size	Purpose	Accuracy
[86]	Long span bridge	5 × 5, 3 × 3, 3 × 3, 3 × 3 (4 layers)	2 × 2, 2 × 2. 2 × 2 (3 layers)	Softmax	28,272 samples	Purpose a data anomaly detection method based on CNN combined with statistical features	94.26%
[87]	steel bridges	3 × 3 (2 layers)	4 × 4 (2 layers)	ReLU	45,645 images	A vision-based method for crack detection in gusset plate welded joints of steel bridges.	98%
[88]	Concrete box girder bridge	3 × 3 × 16, 3 × 3 × 32, 3 × 3 × 64 (3 layers)	3 layers	ReLU, Softmax	3000 images	Damage detection in girder bridges using modal curvatures gapped smoothing method (GSM) and Convolutional Neural Network	82%.
[89]	Steel girder bridge	5 × 1 × 2 (5 layers)	2 (5 layers)	ReLU	Each dataset has 2800 samples (3 datasets)	Proposes a new intelligent damage diagnosis method for bridges called sub-domain adaptive deep transfer learning network (SADTLN), to solve the feature generalization problem in different bridges.	Above 98%
[90]	Steel girder bridge	5 × 1 × 2 (5 layers)	2 (5 layers)	ReLU	Each dataset has 2800 samples (3 datasets)	Proposes a new open-set deep transfer learning algorithm based on joint weighted sub-domain adaptation.	above 94%.
[91]	Steel railway bridge	7 layers 64 × (1100), 32 × (1,3), 128 × (1,3), 32 × (1,3), 128 × (1,3), 8 × (1,3), 32 × (1,3)	N/A	Softmax, ReLU	1300 samples	Damage classification of in-service steel railway bridges using a novel vibration-based convolutional neural network	Accuracy approaching 100%
[92]	high-speed rail (HSR) reinforced concrete (RC) bridges	5 layers 227 × 227 × 3 27 × 27 × 96 13 × 13 × 256 13 × 13 × 348 13 × 13 × 348	3 layers 55 × 55 × 96 27 × 27 × 256 13 × 13 × 256	ReLU	6600 samples	Convolutional neural networks (CNNs)-based multi-category damage detection and recognition using test images	86% for cracks, 82% for reinforcement exposure 70% for concrete spalling,
[93]	cable-stayed bridge	3 layers 10 × 10	3 layers 6 × 6	Softmax	17,856 samples	Deploy deep learning to investigate a loss factor function (LF) for measuring energy dispersal.	96.15%
[94]	long-span cable-stayed bridge	96 × 96 × 8 44 × 44 × 16 18 × 18 × 32 (3 layers)	48 × 48 × 8 22 × 22 × 16 9 × 9 × 32 (3 layers)	ReLU, Softmax	21,000 images	Proposes a hyperparameter-tuned convolutional neural network (CNN) for multiclass imbalanced anomaly detection (CNN-MIAD) modeling.	97.74%.
[95]	Reinforced concrete bridge	2 convolution layers for load data (47 × 71) 3 convolution layers for strain data (2 × 5)	Not specified	ReLU	13,000 samples	Damage localization approach using a convolutional neural network (CNN).	87.3%
[96]	Steel truss bridge	448, 160, 16 (3 layers)	224, 80, 8 (3 layers)	Hyperbolic tangent	81,920 samples	Proposes a methodology based on the autocorrelation of vibration signals to detect, locate and quantify corrosion damage.	97%,
[97]	long-span steel bridge	32 × 32 × 16, 16 × 16 × 32, 8 × 8 × 64 4 × 4 × 128 (4 layers)	16 × 16 × 16, 8 × 8 × 32, 4 × 4 × 64 1 × 1 × 128 (4 layers)	ReLU, Softmax	15,708 samples of size 32 × 32 pixels	Proposes a novel approach for crack recognition in digital images.	Not specified
[98]	concrete bridge decks	3 layers	(3 layers)	ReLU, Softmax	4000 GPR images	An automated rebar localization and detection algorithm.	99.60% ± 0.85%.
[100]	steel truss railway bridges	3 × 3 × 3(3 layers)	N/A	ReLU	500 Epochs	A method of locating damage in bridges through classification tools, enabling automatic analysis of raw strain sensor signals without any pre-processing or preliminary feature extraction.	93%
[101]	pedestrian bridge	(1 layer)	(1 layer)	ReLU, Softmax	488 observations.	Estimation of crowd flow and load on pedestrian bridges based on a novel combination of sensing technologies that include the employment of structurally mounted Fiber Bragg Grating (FBG) Fiber Optic Sensors (FOSs) in conjunction with individually held wearable sensing devices incorporating Inertial Measurement Units (IMUs).	peak testing accuracy for single-class motion speed classification at 98%, multi-class motion speed load characterization classification at 91%, and percentage error for load estimation regression reaching a minimum of 9%.
[102]	Steel bridge	CNN1: 5 × 1 (3 layers) CNN2: 10 × 1 (3 layers)	CNN1: 5 × 1 (3 layers) CNN2: 5 × 1 (3 layers)	ReLU	151,238 instances	Proposes an end-to-end framework to evaluate the health of bridges by exploring objective features and correlations of multiple monitoring factors.	F1 score, precision and recall, specificity (SPC), G-Mean (G-M) and F-Measure (F-M) above 0.93 for all above matrices

## Data Availability

Not applicable.

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
