# Peer review of "Algorithms and Techniques for the Structural Health Monitoring of Bridges: Systematic Literature Review"

_sensors, 2023, doi:10.3390/s23094230_

Round 1

Reviewer 1 Report

This study presents the applications of machine learning algorithms used for pattern recognition process for SHM in bridges. To accomplish this objective, SLR has been performed to recognize 45 research articles. On the basis of different machine learning techniques, selected researches are classified into three different categories and 6 learning algorithms are discussed.

1. In abstract, please further emphasize the innovation of the paper

2. This study presents many machine learning algorithms for SHM, including three different categories and 6 learning algorithms, however, the Bayesian method is a very important algorithm for SHM data modelling. However, However, the application of Bayesian method in structural health monitoring is rarely discussed in this paper. The following papers on Bayesian algorithms are recommended for the references.

On the application of kernelised Bayesian transfer learning to population-based structural health monitoring[J]. Mechanical Systems and Signal Processing, 2022, 167:108519

Modelling and forecasting of SHM strain measurement for a large-scale suspension bridge during typhoon events using variational heteroscedasic Gaussian process, Engineering Structures, 2021, 251, 113554.
A variational Bayesian neural network for structural health monitoring and cost-informed decision-making in miter gates:[J]. Structural Health Monitoring, 2022, 21(1):4-18.   Structural system reliability analysis based on improved explicit connectivity BNs, KSCE Journal of Civil Engineering, 2018. 3.12, 22(3): 916~927.

3. Why only paper published in IEEE, SPRINGER. ELSEVIER are selected for the review?   This will affect the comprehensiveness and systematicness of the review.

4. Table 7 and Table 8: Some numbers in the table are centered, some are left aligned, and the format is not uniform.

5. The format of Table 13 is messy. There are some typos in Table 5.

6. The reference format is not uniform, e.g., [11], [12], [20].

7. Table 9 summarizes only the three bridge types, which seem not sufficient. There are other common bridge types, such as suspension Bridges, long span rigid frame bridges.

8. In conclusions, please add specific literature summary results. In the current form, the conclusions only presented what the authors had done, however, the conclusions don't present the final findings.

Minor editing of English language required.

Reviewer 2 Report

The paper presents a review on ML techniques in SHM. The paper, is well written and structured but it needs of some improvements. Here my comments: 

 - From Table 1 it seems that only 6 papers present the contents of the review. I believe that this table should be shifted later in the paper, by better motivating the aim of the table.

- Honestly Section 2 and all subsections could be abbreviated. No interesting information about the topic of the review is present 

- Talking about ML, the review misses of XAI methods and deep learning method for monitoring the health state of bridges, as one of the main current topics nowadays developed. I suggest to check one of the last research about the topic and the references therein (10.1016/j.engfailanal.2023.107237)

- Some specific information (e.g., formulations) should be added for the methodology reported (i.e., random forests or decision trees). This can value the review

- Section 5 should be not reported as question and answer, but like a discussion

- In the end, I suggest to highlight what are the future steps and the direction in which the reasearch should go in this topic

Round 2

Reviewer 1 Report

Generally, the reviewer’s concern is addressed accordingly. There is no further comments.

Reviewer 2 Report

The Reviewer's comments have been addressed. Hence, the paper is ready to be published.